# From Soil to Stream: Modeling the Catchment-Scale Hydrological Effects of Increased Soil Organic Carbon

Malve Heinz<sup>1,2,3</sup>, Annelie Holzkämper<sup>1,2</sup>, Rohini Kumar<sup>4</sup>, Sélène Ledain<sup>2</sup>, Pascal Horton<sup>1,3</sup>, and Bettina Schaefli<sup>1,3</sup>

**Correspondence:** Malve Heinz (malve.heinz@unibe.ch)

**Abstract.** Droughts are increasingly threatening agricultural productivity. One potential adaptation is to increase the soil water retention capacity, which can be achieved by enhancing soil organic carbon (SOC) through agricultural management. We investigated how increasing SOC affects catchment-scale hydrology including extremes. SOC increases were implemented via adjustments to soil hydraulic parameters ( $\rho_b$ ,  $\theta_{PWP}$ ,  $\theta_{FC}$ ,  $\theta_{Sat}$ ,  $K_{sat}$ ) in a mesoscale hydrologic modeling (mHM) framework, following literature-reported effects. Our analysis focuses on the medium-sized, agriculturally dominated Broye catchment in Western Switzerland, wherein we evaluated five SOC increase scenarios of varying depth and magnitude. At the plot scale, SOC increases resulted in higher net soil water content (2.99–8.13%) and slightly higher evapotranspiration (0.15–0.4%), while subsurface runoff was reduced (0.28-0.72% across all scenarios). These values represent overall net changes; while at shorter timescales, the magnitude and even direction of effects varied by season and location. Increased water retention meant more soil water was retained and latter evaporated and less was available for groundwater recharge and eventually as streamflow. At the catchment scale, streamflows were slightly reduced, with peak flows modestly attenuated. Low flow responses depended on catchment characteristics and timing. In warmer and drier subcatchments, low flow frequency increased in some years, whereas in cooler and wetter subcatchments, conditions in spring and early summer produced a beneficial effect, slightly reducing low flow frequency. Overall our analysis suggest that a large-scale increase in SOC, while benefiting agricultural productivity and peak flow attenuation, may also induce trade-offs by potentially reducing groundwater recharge and downstream water availability.

**Keywords.** SOC increase; drought; land use change, climate change adaptation; mHM; agro-hydrological modeling; Switzerland

#### 1 Introduction

Agricultural productivity is strongly influenced by hydro-climatic variability. Meteorological, hydrological, and soil moisture droughts can co-occur and substantially reduce crop yields (Hou et al., 2024; Tijdeman et al., 2022). Compared to soils under natural vegetation, agricultural soils are more prone to soil moisture depletion (Yu et al., 2019). This heightened vulnerability

<sup>&</sup>lt;sup>1</sup>Oeschger Centre for Climate Change Research, University of Bern, Switzerland

<sup>&</sup>lt;sup>2</sup>Agroecology and Environment, Agroscope, Switzerland

<sup>&</sup>lt;sup>3</sup>Institute of Geography, University of Bern, Switzerland

<sup>&</sup>lt;sup>4</sup>Department of Computational Hydrosystems, Helmholtz Centre for Environmental Research - UFZ, Leipzig, Germany

35

is also a consequence of long-term soil degradation: i.e., intensive management has depleted soil organic carbon (Córdova et al., 2025), heavy machinery has compacted soils, increased surface runoff and reduced hydraulic conductivity and water retention (Keller et al., 2019), and bare fallow practices have promoted erosion and weakened soil structure (Poeplau and Don, 2015). Such degradation processes reduce the capacity of soils to buffer hydro-climatic extremes, thereby amplifying flood and drought impacts (Saco et al., 2021). With climate change projected to intensify hydro-climatic variability, soil moisture droughts in Europe are expected to expand in both area and duration (Samaniego et al., 2018).

To understand how climate change impacts agriculture, it is crucial to consider both the responses of plants and those of farmers. When precipitation and soil moisture deficits coincide with high air temperatures and high evaporative demand, plants close their stomata to limit water loss (Gupta et al., 2020). Depending on the timing and duration of a drought, plant survival rate, growth, and yield quantity and quality can be adversely affected (Dietz et al., 2021). Irrigation has traditionally been the main option to reduce drought stress and mitigate yield loss, even in water-rich countries like Switzerland (Wriedt et al., 2009; Baumgartner et al., 2025).

However, irrigation increasingly competes with ecological needs and other sectors (Brunner et al., 2019). In Switzerland, water withdrawals from rivers may be restricted when low flows reach critical thresholds for aquatic ecosystems (Heinz et al., 2025), making yield losses unavoidable. While projections for larger Central European catchments show mixed trends (Marx et al., 2018), summer low flows in lowland Swiss catchments like the Broye are projected to become more frequent and severe from mid-century onwards under climate change (Muelchi et al., 2021a). Hence, irrigation restrictions will probably become more frequent in the future. To increase the resilience of agricultural cropping systems to droughts, management can be adapted to enhance the soil's function as a hydrological buffer, particularly its water retention capacity (Hou et al., 2024).

A key metric for this is plant available water capacity (PAWC), defined as the difference in volumetric soil moisture between field capacity ( $\theta_{FC}$ ) and the permanent wilting point ( $\theta_{PWP}$ ). PAWC thus represents the range of soil water that can potentially be accessed by plants. The actual plant available water (PAW) is the portion of PAWC currently present in the soil, representing the water that plants can actually extract at a given time. Other parameters that are key to assessing the soil's hydraulic behavior are the bulk density ( $\rho_b$ ) and saturated hydraulic conductivity ( $K_{sat}$ ).  $\rho_b$  is defined as the dry mass relative to total soil volume (commonly in  $1.2\,\mathrm{g\,cm^{-3}}$ ), while  $K_{sat}$  describes the rate at which water flows through saturated soil (cm d<sup>-1</sup>).

Agricultural practices that enhance soil structure and increase organic matter, such as conservational tillage, organic amendments, or cover cropping, can influence these parameters, especially  $\rho_b$ ,  $K_{\rm sat}$ , and ultimately PAWC (Lal, 2004; Blanco-Canqui et al., 2009; Blanco-Canqui et al., 2023; Chalise et al., 2019; Bormann et al., 2007). For example, applying organic amendments enhances microbial activity and organic matter decomposition. This promotes soil aggregation, which generally reduces  $\rho_b$ , increases porosity and pore connectivity, and thereby improves infiltration, water retention, and ultimately  $K_{\rm sat}$ . Shi et al. (2016) observed these effects in a silt loam, but the magnitude and direction of the changes can vary depending on soil texture and structure. In addition to enhancing water retention, increasing soil organic carbon (SOC) offers a co-benefit of contributing to negative  $CO_2$  emissions through carbon sequestration, particularly in the relatively undisturbed subsoil (Button et al., 2022). (Heinz et al., 2025) showed that increasing SOC in potato fields can reduce drought stress and yield losses for a case study in

75

80

Switzerland. Assuming that such adaptive management is scaled up and applied on a larger area, raises the question on how do these field-level interventions affect catchment-scale hydrological processes?

We know that hydrologic processes at the catchment scale can be influenced by local-scale changes (Öztürk et al., 2013; Ni et al., 2021). In recent years, the use of natural processes to manage water in the landscape, often referred to as nature-based solutions, has received increasing attention (Collentine and Futter, 2018; Vann et al., 2025). These practices include structural measures such as wetland and floodplain restoration, afforestation, riparian buffer strips, or terracing (Deng et al., 2021; Potter, 1991; Krois and Schulte, 2012), as well as targeted arable soil management (Vann et al., 2025). Terracing has been investigated through both modeling and field-based studies, showing potential to enhance soil moisture and reduce erosion locally, while possibly limiting downstream water availability (Deng et al., 2021). Soil conservation practices, such as conservational tillage and gully treatment, have been evaluated in data-based case studies and can decrease flood peaks and increase flood rise times (Potter, 1991). Modeling studies also indicate that practices like no-tillage can reduce hydraulic conductivity, leading to higher runoff and peak flows (Krois and Schulte, 2012; Moussa et al., 2002). Similarly, Fatichi et al. (2014) used the mechanistic model Tethys-Chloris (Fatichi et al., 2012) to analyze grassland management effects (e.g., grazing, mowing, compaction) from plot to catchment scale. They found that detectable catchment-scale impacts generally require large interventions or long observation periods. Fatichi et al. (2012) highlight that modeling is often the only feasible way to assess such effects, given data limitations and the need for comparable catchments with and without management adaptations. To our knowledge, the impacts of field-scale agricultural management practices designed to enhance soil water retention on evaporation, groundwater recharge, and hydrological extremes have not yet been systematically explored.

To address this gap, we investigate how increasing soil organic carbon (SOC) affects catchment-scale hydrology, including hydrologic extremes such as low and peak flows. A model-based approach is used because long-term observational data capturing pre- and post-management hydrological conditions are not available. We hypothesize that increasing SOC, and thus soil water retention, alters the timing and partitioning of water fluxes – potentially mitigating low-flow conditions by sustaining soil moisture and discharge during dry periods, while modestly reducing peak flows through enhanced retention capacity.

For this analysis, we implement the distributed mesoscale hydrological model mHM (Samaniego et al., 2010; Kumar et al., 2013), representing agricultural management as scenarios of varying SOC increases. Changes in SOC propagate through the model via adjusted soil hydraulic parameters ( $\rho_b$ ,  $\theta_{PWP}$ ,  $\theta_{FC}$ ,  $\theta_{Sat}$ ,  $K_{sat}$ ), reflecting observed SOC effects reported in the literature. The selected case study is the lowland, mid-sized agricultural Broye catchment in Western Switzerland, which is prone to agricultural droughts, summer low flows and has a good data coverage. We chose mHM for this analysis, as it is a distributed, open-source model under active development with a growing user community (https://mhm-ufz.org). The model has been successfully used to simulate not only discharge, but also the spatiotemporal dynamics of runoff, evapotranspiration, and soil moisture across diverse European catchments (Samaniego et al., 2010; Kumar et al., 2013; Samaniego et al., 2016). mHM has also been applied to generate soil moisture time series for drought analysis and serves as the basis for the German drought monitor (Thober et al., 2015; Samaniego et al., 2018; Boeing et al., 2025).

#### 90 2 Methods

110

Our analysis framework is based on the catchment-scale hydrological model mHM, duly calibrated and evaluated using observed discharge time series (Sect. 4.1). Based on the reviewed literature (Sect. 2.1), we implement the effects of a theoretical combination of agricultural management practices for several scenarios of soil organic carbon (SOC) increases (Sect. 3.4). These scenarios are implemented by adjusting the input data for bulk density ( $\rho_b$ ) using a pedotransfer function that considers SOC. The pedotransfer function used internally in mHM to calculate saturated hydraulic conductivity ( $K_{sat}$ ) is also adjusted to consider SOC (see Sect. 2.3). We evaluate the effect of different SOC increase scenarios on the effective model parameters, hydrological states and fluxes at the grid scale and their effect on discharge including hydrological extremes for each sub-catchment.

# 2.1 Literature-informed adjustment of soil hydraulic parameters

We conducted a literature review to identify studies that examined changes in soil properties resulting from management adaptations aimed at increasing SOC. Estimates of soil hydraulic properties—such as soil moisture at field capacity ( $\theta_{FC}$ ), permanent wilting point ( $\theta_{PWP}$ ), or saturated hydraulic conductivity ( $K_{sat}$ ), derived from pedotransfer functions (PTFs) can vary considerably and are a source of uncertainty (Paschalis et al., 2022; Turek et al., 2025). However, since PTFs are (ideally) trained on large soil datasets from similar pedoclimatic conditions, they should support a broad generalization and enable the prediction of difficult-to-measure parameters from more easily observable ones. Moreover, they typically cover a wider range of soil textures than field or experimental studies. In Table 1, we distinguish between the reported orders of magnitude from experimental and modeling studies, both of which we consider in our analysis.

A range of management practices are reported that increase SOC: including cover cropping, diversified crop rotations, the application of organic amendments (e.g., compost or manure), the retention of crop residues and the application of biochar (Table 1). These practices are often combined, and the magnitude of SOC increase varies depending on site-specific conditions, depth, and implementation duration. The reported increases in SOC range from 7% to 36%, 20% to 220%, and 60% to absolute increases of approximately + 1% by mass (Haruna et al., 2020; Hao et al., 2023; Blanco-Canqui et al., 2023; Shi et al., 2016; Blanco-Canqui et al., 2009).

In addition to changes in SOC, several studies report concurrent effects on other soil hydraulic properties. A reduction in bulk density ( $\rho_b$ ) is frequently observed; the effect varies from -1% to -4% through cover cropping (Haruna et al., 2020; Hao et al., 2023) to -14% for a silt loam in response to long-term organic amendments (Shi et al., 2016).

The ranges of change in saturated hydraulic conductivity ( $K_{\rm sat}$ ) are particularly variable, with reported increases of 50% to 250%, 40% to 360% and 95% depending on the practice and the site.  $K_{\rm sat}$  underlies large variability and is a generally hard-to-measure and even harder-to-estimate variable and should be handled with caution (Verrelst et al., 2019).

**Table 1.** Summary of soil property changes under different practices and modeling studies. PAWC = plant available water capacity (range between  $\theta_{PWP}$  and  $\theta_{FC}$ ).

SOM converted to SOC assuming SOM  $\approx 58\%$  SOC.

| Stud          | Study Practice covered                                                                    | Soil texture                         | \ \rangle SOC                                      | \ \ \ \ \ \ \ \ \ \ \ \ \ \ \ \ \ \ \ | $\Delta K_{ m cat}$                                       | θ Δ                                                                                                                                                                       | Source                                                       |
|---------------|-------------------------------------------------------------------------------------------|--------------------------------------|----------------------------------------------------|---------------------------------------|-----------------------------------------------------------|---------------------------------------------------------------------------------------------------------------------------------------------------------------------------|--------------------------------------------------------------|
| type          |                                                                                           |                                      |                                                    |                                       | 1185                                                      |                                                                                                                                                                           |                                                              |
|               | Cover cropping                                                                            | various                              | +7-36%                                             | -1 to -4%                             | +40-360%                                                  | +4–20% PAW                                                                                                                                                                | Haruna et al. (2020);<br>Hao et al. (2023)                   |
|               | Organic farming (diverse crop rotations, organic amendments, manure application, tillage) | various                              | +20% to<br>+220%<br>(*1.2-*3.2)                    | -2 to                                 | +50-250%                                                  | +4–54% PAW                                                                                                                                                                | Blanco-Canqui et al. (2023)                                  |
| Μ             | (Long-term) organic manure application                                                    | Silt loam                            | +60% (topsoil)                                     | -14%<br>(topsoil)                     | Likely ↑ but<br>not significant                           | +7.8–9.7% θ <sub>sat</sub>                                                                                                                                                | Shi et al. (2016)                                            |
| y or revie    | Organic manure + biochar application                                                      | Sandy loam                           | Likely ↑                                           | -2.6 to                               | +25%                                                      | Likely ↑                                                                                                                                                                  | Veettil et al. (2024)                                        |
| periment stud | Review on crop residue effect                                                             | Silt loam<br>Clay loam               | +1% (mass)<br>+0.65% (mass)                        | -3%                                   | %06+<br>%96+                                              | +33% PAWC<br>+65% PAWC                                                                                                                                                    | Blanco-Canqui et al. (2009)                                  |
| хә            | Analysis of soil databases on SOC effects on soil hydraulic properties                    | Various<br>textures (US)             | +0.6% (mass)<br>+1% (mass)<br>+1.2% (mass)         |                                       |                                                           | +1.5–1.7% PAWC<br>+2–5% PAWC<br>up to +50% PAWC                                                                                                                           | Libohova et al. (2018) Archer and David (2005) Hudson (1994) |
|               | Analysis of effect of SOC on soil properties (high clay content reduces impact)           | Coarse (Sand 50%) Fine (Sand 20%)    | 1% vs. 3%; 3%<br>vs. 5%<br>1% vs. 3%; 3%<br>vs. 5% |                                       | +20% to +85%,<br>+0 to +7%<br>+14% to +17%,<br>+0 to +25% |                                                                                                                                                                           | Rawls et al. (2004)                                          |
| ybuts gnile   | Modeling impacts of varying $\rho_b$ on soil hydraulic properties                         | Clay loam<br>Silt loam<br>Sandy loam |                                                    | -10%<br>-10%<br>-10%                  | +127%<br>+114%<br>+86.4%                                  | +7% $\theta_{\text{sat}}$ , -8% $\theta_{\text{res}}$<br>+7% $\theta_{\text{sat}}$ , -8.7% $\theta_{\text{res}}$<br>+7% $\theta_{\text{sat}}$ , -7% $\theta_{\text{res}}$ | Kojima et al. (2018)                                         |
| ppou          | Modeling land use change scenarios                                                        | various                              |                                                    | -5%,<br>-10%,<br>-15%                 | +35%, +70%,<br>+140%                                      | +5%, +20%, +30%<br>PAWC                                                                                                                                                   | Bormann et al. (2007)                                        |

125

145

Soil moisture ( $\theta$ ) and especially plant available water capacity (PAWC), are reported to increase in the range of 4% to 20%, 4% to 54%, 33% and 65%, following increasing SOC and decreasing  $\rho_b$  (Haruna et al., 2020; Hao et al., 2023; Blanco-Canqui et al., 2009).

In Blanco-Canqui et al. (2009), reducing crop residue cover from 100% to 0% decreased SOC, increased  $\rho_b$ , and reduced  $K_{\text{sat}}$  and PAWC. In Table 1, we assume that increasing residue cover from 0% to 100% would have the opposite effects: increasing SOC, reducing BD, and increasing  $K_{\text{sat}}$  and PAWC.

The role of tillage is more complex. While reduced or no-tillage is often associated with higher SOC in the topsoil, it primarily leads to a redistribution of organic matter, with less SOC in deeper layers, and total SOC differences are not always significant (Bragazza et al., 2025). Tillage is often used in organic farming to control weeds, which can offset some of the beneficial effects of organic farming practices. In particular, Blanco-Canqui et al. (2023) describe how tillage can negatively affect bulk density ( $\rho_b$ ) and  $K_{sat}$ , potentially counteracting the positive impacts of increased SOC in organic management systems, depending on tillage frequency and intensity.

# 2.2 Model description

The mesoscale Hydrologic Model (mHM, v. 5.13.1; https://mhm-ufz.org) is a spatially distributed, process-based model designed to simulate major hydrological processes and water balance across diverse hydroclimatic regions and scales (Samaniego et al., 2010; Kumar et al., 2013; Feigl et al., 2022). The computation of soil moisture processes and the generation of mobile water takes place at a grid scale, followed by a HBV-like soil moisture-runoff transformation to transform grid-scale mobile water to grid-scale runoff, followed by transfer and routing from grid cell to grid cell following topography-based flow directions (see below). The multiscale-parameter regionalization (MPR) is a key feature of mHM, which allows for both high-resolution spatial input data and computational efficiency (Samaniego et al., 2010; Kumar et al., 2013). Using transfer functions, effective model parameters (such as hydraulic conductivity) at the modelling grid-scale are estimated from spatial input parameters such as soil texture. These effective parameters are then internally upscaled to the (coarser) model resolution using different operators such as harmonic or arithmetic mean, while retaining spatial variability (Samaniego et al., 2010; Kumar et al., 2013). More detailed descriptions are available in the work of Samaniego et al. (2010); Kumar et al. (2013) with more specific details on soil hydraulic parameterizations in Livneh et al. (2015).

The main processes simulated in mHM are canopy interception, snow accumulation and melt, evapotranspiration, infiltration, soil moisture storage, surface runoff, lateral subsurface flow (called interflow in mHM), percolation, groundwater storage, baseflow and in-stream routing (Samaniego et al., 2010) (Fig. 1). Snow accumulation is simulated with a simple temperature threshold; snowmelt is based on a degree-day method. In mHM, surface runoff can only occur on (nearly) impervious grid cells representing sealed areas such as streets or buildings. Potential runoff from excess water is assumed to re-infiltrate at the grid-scale and is, therefore, not simulated as a separate process in mHM. This is justified by the typically recommended grid resolution of 1 km to 50 km (Samaniego et al., 2010).

The soil moisture and runoff generation schemes in mHM are conceptually based on the HBV model (Bergström, 1995), with some differences: mHM simulates soil moisture dynamics per soil layer (HBV usually has a single layer); the routine

Figure 1. mHM flowdiagram adapted from Kumar et al. (2013)

160

170

175

is described in more detail in Sect. 2.3. Once mobile water is generated per soil layer, the HBV conceptualization is used to transform grid-scale mobile water into grid-scale runoff. Each grid cell uses two subsurface storages fed with the sum of the mobile water from the soil moisture routine. The upper unsaturated storage generates faster responding interflow and the other slower responding baseflow (Fig. 1). Fast interflow occurs only if the water level in the storage zone exceeds a threshold; slow interflow is a permanent flux governed by the water level in the first bucket (Livneh et al., 2015). The remaining water level in this zone is the base for the percolation flux, encoded as a linear function of the water level. The percolation feeds the deeper saturated zone, supposed to emulate groundwater storage, where baseflow is again parameterized as a linear function of the water level (Samaniego et al., 2010).

The total generated runoff (interflows and baseflow) from each grid cell is routed through the modelling domain by the multiscale Routing Model (mRM), a key component of the model (Thober et al., 2019). Grid-scale runoff is transferred from cell to cell following topography-based flow direction and flow accumulation map. The routing algorithm applies the kinematic wave equation with spatially varying flow celerity parameterized by slope (Thober et al., 2019). An adaptive time-stepping scheme is used to ensure numerical stability across resolutions. Shrestha et al. (2025) developed the subgrid catchment conservation (SCC) routine specifically for mHM as an alternative to the commonly used D8 algorithm (O'Callaghan and Mark, 1984). This approach addresses the catchment size problem that arises when small catchments are simulated at coarse resolution, which can lead to over- or underestimation of catchment area and the resulting streamflow. For cells intersecting several subcatchments, SCC allows water to partition into different neighboring cells. Due to this study's relatively small catchment size, we also employ the SCC algorithm, which reduces biases in discharge between different sub-catchments (Shrestha et al., 2025).

In the configuration chosen for this study (see Sect. 3.3), the model has 47 (global) parameters that are calibrated based on observed streamflow (calibrated parameter values in Table A3). mHM has a built-in calibration algorithm based on a dynamically dimensioned search algorithm (Tolson and Shoemaker, 2007) for single objective parameter optimization. The users can choose between several performance criteria (https://mhm-ufz.org). The retained calibration options for the case studies at hand are further discussed in Sect. 3.3. The number of iterations is set to 2500, which has been successfully used to calibrate the mHM model in other studies (Kumar et al., 2010; Samaniego et al., 2017; Shrestha et al., 2024).

#### 2.3 Parameterization of mHM soil moisture dynamics related to SOC changes

The mHM model represents root-zone soil moisture dynamics across multiple soil layers, with each layer corresponding to an individual soil water reservoir. The water balance within each reservoir is primarily controlled by incoming fluxes – snowmelt and rainfall in the uppermost layer, or percolation from the overlying soil layer in lower layers – and outgoing fluxes, including downward percolation and layer-specific evapotranspiration. Each soil layer has an upper soil water limit, represented by  $\theta_{\text{sat}}$ , which acts as a threshold for storage capacity.  $\theta_{\text{sat}}$  is estimated using the PTF by Zacharias and Wessolek (2007):

$$\theta_{\text{sat}} = C_{\text{constant}} + C_{\text{clav}} \cdot \tau_{\text{clay}} + C_{\text{DB}} \cdot \rho_b \tag{1}$$

Figure 2. Adjustment of bulk density ( $\rho_b$ ) for organic matter ( $p_{OM}$ ) in the default mHM routine, compared to the model version adapted for this study using spatially distributed SOC data.

where  $C_{\text{clay}}$  is the clay content, and  $C_{\text{constant}}$ ,  $\tau_{\text{clay}}$ , and  $C_{\text{DB}}$  are (global) parameters that are calibrated (listed in Table A3). At each time step, the current water content  $\theta$  in each soil layer is compared to  $\theta_{\text{sat}}$ ; if  $\theta$  is below saturation, infiltration into the layer is allowed. A portion of the incoming water is retained in the current layer, while the remainder percolates into the next layer (see Equations in Appendix A). This also means that if  $\theta_{\text{sat}}$  (i.e., the soil's water retention) increases, then for the same water input, less water infiltrates to deeper layers.

In the default mHM setup, bulk density ( $\rho_b$ ) is internally estimated from a user-defined mineral bulk density ( $\rho_{b,\text{min}}$ ) and modified using an organic matter parameter ( $p_{\text{OM}}$ ), which can be fixed or calibrated but is spatially uniform (Fig. 2). Saturated hydraulic conductivity ( $K_{\text{sat}}$ ) in mHM is derived using the pedotransfer function (PTF) from Cosby et al. (1984), based on sand and clay content. We modify this parameterization to evaluate the effect of different SOC scenarios by directly linking SOC to  $\rho_b$  and  $K_{\text{sat}}$  (Fig. 2). Specifically, we bypass the internal  $p_{\text{OM}}$  routine and instead input SOC-adjusted  $\rho_b$  values directly, using the PTF from Manrique and Jones (1991), adapted by De Vos et al. (2005):

$$\rho_b = 1.660 - 0.318\sqrt{\tau_{\text{SOC}}},$$
(2)

where  $\tau_{SOC}$  is the SOC content. Here we follow Minasny and McBratney (2018) who showed that SOC consistently affects  $\rho_b$  in a largely texture-independent way. By representing SOC changes through  $\rho_b$  in pedotransfer functions, the resulting soil hydraulic parameters naturally reflect SOC effects (Zacharias and Wessolek, 2007).

Above mentioned adaptation also allows us to capture the observed relationship between increasing SOC and decreasing  $\rho_b$ , which is generally linked to higher  $K_{\text{sat}}$  (Saxton and Rawls, 2006). To incorporate the effect of increasing SOC onto  $K_{\text{sat}}$ , we also replace the default PTF with the one proposed by Vereecken et al. (1990), as listed in Lee (2005), which includes SOC and  $\rho_b$  as predictors:

$$K_{\text{sat}} = C_{\text{Ksat1}} \exp\left(C_{\text{Ksat2}} - C_{\text{Ksat3}} \ln(\tau_{\text{clay}}) - C_{\text{Ksat4}} \ln(\tau_{\text{sand}}) - C_{\text{Ksat5}} \ln(\tau_{\text{SOC}}) - C_{\text{Ksat6}} \rho_b\right),\tag{3}$$

210

where  $\tau_{\text{clay}}$  and  $\tau_{\text{sand}}$  are the clay and the sand content, the parameters  $C_{\text{Ksat1}}$  to  $C_{\text{Ksat6}}$  are constants (listed in Appendix A). In mHM,  $\theta_{\text{FC}}$  is parameterized as a function of  $K_{\text{sat}}$ , such that higher  $K_{\text{sat}}$  corresponds to lower  $\theta_{\text{FC}}$ , based on the PTF derived from soil database analysis by Twarakavi et al. (2009).

These parameter adjustments propagate through the process simulation chain, and their effects on parameters, variables, states, and fluxes in response to increased SOC will be described and illustrated in the results Section (Fig. 6). They influence the estimation of the van Genuchten parameters used to compute  $\theta_{\text{sat}}$ ,  $\alpha$ , n, and m, as well as field capacity ( $\theta_{\text{FC}}$ ) and the permanent wilting point ( $\theta_{\text{PWP}}$ ) (Equations listed in Appendix A). These, in turn, affect the simulated soil moisture ( $\theta$ ) and the associated fluxes, including infiltration, evapotranspiration (ET), lateral subsurface flow, and percolation.

ET in mHM is computed as a reduction of potential evapotranspiration (PET) by a soil moisture stress factor, following the formulation of Feddes et al. (1976) or Jarvis (1989). In this study, we used the mHM process representation of Demirel et al. (2018), which combines the Jarvis approach with a root distribution model based on Jackson et al. (1996). In this configuration, root density varies spatially and vertically as a function of soil field capacity ( $\theta_{FC}$ ).

The reduction from PET, after accounting for canopy interception, to ET is expressed as:

$$ET = PET \cdot f \tag{4}$$

where f is a soil moisture stress function defined by:

Here,  $t_{jarvis}$  is a calibrated threshold parameter,  $\bar{\theta}$  is the normalized soil water content:

$$\overline{\theta} = \frac{\theta - \theta_{\text{pwp}}}{\theta_{\text{sat}} - \theta_{\text{pwp}}},\tag{6}$$

and R is the fraction of roots in each soil layer:

$$R = \left(1 - R_{\text{CoeffFC}}^{d_{\text{u}}}\right) - \left(1 - R_{\text{CoeffFC}}^{d_{\text{l}}}\right) \tag{7}$$

with  $R_{\text{CoeffFC}}$  representing the root fraction coefficient for the layer, and  $d_{\text{u}}$  and  $d_{\text{l}}$  denoting the upper and lower soil layer boundaries (Appendix A). This formulation allows soil-layer specific root fractions to modulate ET in response to soil moisture.

# 3 Data

# 3.1 Study area

We apply the mHM model to the mid-sized (602 km<sup>2</sup>), lowland, pluvial Broye catchment in Western Switzerland (Fig. 3).

The modeling period is constrained by the availability of leaf area index input data and is therefore set to 2015–2022. The year 2015 is used as a warm-up period and is therefore discarded from any analysis. Despite the relatively short study period,

there is considerable variability, with 2018 and 2022 as hot and dry years, 2016 and 2021 as cool and wet years, and some intermediate, less extreme years (2017, 2019 and 2020, Fig. 3). There is no gauging station at the outlet, but there are stations for four subcatchments, whose properties are listed in Fig. 3.

# 235 3.2 Input data

The required input data and their sources are listed in Table 2. The morphological and land use input data have a resolution of 50 m x 50 m, and the meteorological data of 1 km x 1 km, which is also the internal modeling resolution. The water transfer and routing in the model are based on the provided flow direction. However, because the water flow in the flat part of the catchment is not well constrained by the DEM, a reconditioned DEM consistent with the mapped rivers must be calculated. After trying different tools that provided unsatisfactory results, we developed a new tool to seamlessly align DEMs with mapped stream networks, resulting in minimal terrain alteration: hydro-snap (Horton, 2024). The approach is softer than a stream burn-in and alters the DEM only where necessary. It also constrains the flow direction to be consistent with a provided catchment boundary. With the available gridded precipitation data (MeteoSwiss, 2021a), the water balance in the subcatchments Petit Glâne and Arbogne does not close (Appendix 6). The observed annual discharge is far too low compared to the catchment-average precipitation. However, comparable catchments nearby show similarly low discharge values (Canton of Bern, 2025; Canton of Vaud, 2025); accordingly, discharge measurement errors alone cannot explain the difference. The gridded precipitation product we use might well contain interpolation artifacts given the substantial spatial variability of observed precipitation. Therefore, we also explored other precipitation products (Supplementary Material). To reduce potential biases, we eventually combined the gridded precipitation product with data of the nearby meteo stations for the two smaller subcatchments, Arbogne and Petit Glâne (Supplementary Material).

# 3.2.1 LAI

It has been shown that using spatially distributed leaf area index (LAI) instead of monthly look-up tables improves the discharge estimation for the VIC model (Liang, 1994), that mHM is partly based on (Tesemma et al., 2014). Therefore, LAI was inferred from Sentinel-2 imagery using a specifically trained neural network (NN). Sentinel-2 provides multispectral data at up to 10 m resolution with a 3-day revisit time at mid-latitudes. To train the model, a radiative transfer model (PROSAIL; Jacquemoud et al. 2009) was used to simulate vegetation spectral reflectances based on varying leaf and canopy parameters, thereby generating a training database. Here, PROSAIL was parametrised specifically for arable crops in Switzerland.

ESA's Sentinel Application Platform (SNAP) toolbox includes a Biophysical Processor estimating LAI from Sentinel-2 imagery for all vegetation types (Weiss and Baret, 2016). We therefore used a two-model strategy: the generic SNAP model for forests, and a trained neural network for cropland. LAI was predicted at 10 m (NN) and 20 m (SNAP) resolution, then combined using our land-use mask (Table 2, Zanaga et al. 2022). Non-vegetated areas were set to zero. Monthly median values were calculated and upscaled to 50 m resolution using nearest-neighbor interpolation.

**Figure 3. a**: Sub-catchments and gauging station location. **b**: Soil texture and soil organic carbon for the total Broye catchment. **c**: cumulative temperature and precipitation sums for the whole catchment, cumulative sum of discharge for sub-catchments [scaled for easier comparison, in 2022 only data available for the Broye sub-catchment]

**Table 2.** Overview of input data used.

| mHM input data             | Data description & source                                                                            |
|----------------------------|------------------------------------------------------------------------------------------------------|
| Morphological data with a  | 50 m resolution                                                                                      |
| Land use map               | Land use reclassified in three classes: pervious, impervious, forest. Extracted                      |
|                            | from ESA WorldCover (Zanaga et al., 2022).                                                           |
| Soil map                   | Soil type map along with the corresponding table of soil horizons (texture %,                        |
|                            | bulk density g/cm <sup>3</sup> ) (Stumpf et al., 2023).                                              |
| Hydrogeological map        | Map and corresponding table of the main hydrogeological classes (Federal                             |
|                            | Office for the Environment (FOEN), 2009).                                                            |
| Digital Elevevation Model  | DEM reconditionned with hydro-snap (Horton, 2024) and based on the                                   |
|                            | swissALTI3D product (swisstopo, 2021).                                                               |
| Flow direction map         | Flow direction computed by pysheds (Bartos, 2020) on the reconditionned                              |
|                            | DEM.                                                                                                 |
| Flow accumulation map      | Flow accumulation computed from the flow direction map.                                              |
| Aspect map                 | Aspect map computed from the DEM                                                                     |
| Slope map                  | Slope map puted from the DEM                                                                         |
| Gauges position map        | Map with location of gauging stations                                                                |
| Forcing data with a 1000 m | resolution                                                                                           |
| Precipitation              | Daily precipitation (mm/d) from the RhiresD dataset (MeteoSwiss, 2021a)                              |
| Temperature                | Average daily temperature (°C) from the TabsD dataset (MeteoSwiss, 2021b)                            |
| PET                        | Daily PET calculated after Priestley-Taylor (mm/d) using data from swisstopo                         |
|                            | (2021); MeteoSwiss (2021b); Stöckli (2013)                                                           |
| LAI                        | Monthly LAI derived from Sentinel 2 satellite data                                                   |
| Discharge                  | Daily discharge (m <sup>3</sup> s <sup>-1</sup> ) provided by DGE Vaud (2025a, b, c); Federal Office |
|                            | for the Environment (FOEN) (2023)                                                                    |

# 3.3 Model set-up and evaluation

Different options are available to represent the hydrological processes in mHM (see https://mhm-ufz.org for details). We selected the default options (Samaniego et al., 2024), except for the soil moisture and the evapotranspiration routine. For the soil moisture routine, we selected the option in which ET for each soil layer is constrained by relative available soil moisture instead of being uniform for each land use class (following Jarvis (1989) as implemented and documented by Demirel et al. (2018)). This option allows for a spatially varying root fraction distribution depending on the soil's field capacity ( $\theta_{FC}$ ), which is an advantage in the presence of a high-quality soil database of high resolution (90 m x 90 m) (Stumpf et al., 2023). In contrast to most crop and land surface models, where root distribution is prescribed as a depth-dependent function independent of soil moisture (Maan et al., 2023), mHM explicitly links the root distribution to the soil's  $\theta_{FC}$  following Demirel et al. (2018).

We compute PET as an external model input according to the Priestley-Taylor method (Priestley and Taylor, 1972), which uses average temperature, solar radiation and elevation as input. We set the model options such that PET is further distributed in space based on aspect, as implemented by Demirci and Demirel (2023). It should be noted that the mHM option to correct PET based on LAI data led to unrealistically high PET/ET values in our case and was thus not used.

The model is calibrated using as performance criterion the Kling-Gupta efficiency (KGE) (Gupta et al., 2009), calculated on each of the observed streamflow time series (at the four gauges) and averaged thereafter (without weighting). The retained period for calibration is 2016-2019 and for evaluation 2020-2022. The specific calibration setting is the result of manual explorations of different objective functions and of number of iterations. With fitting the model based on the KGE, we could get the overall best performance while maintaining realistic dynamics of all states and fluxes.

We also evaluate the model performance for soil moisture using observed timeseries of volumetric water content at three depths from a grassland site close to the Broye gauging station in Payerne, measured as part of the Swiss Soil Moisture EXperiment SwissSMEX (Mittelbach and Seneviratne, 2012).

# 3.4 SOC change experiments

We apply different scenarios to evaluate the effect of increased SOC on catchment hydrology to i) represent possible outcomes from long-term agricultural management adaptations (Sect. 2.1), and ii) test the model's sensitivity towards different levels and depths of SOC increases (Fig. 4). In the Broye catchment, SOC is on average around 2.2% in the first 30 cm (soil layer 1), and approximately 0.9% between 30 cm and 60 cm (layer 2, Fig. 3). The SOC ratio between layer 2 and layer 1 is therefore approximately 60 %. We apply this depth-decrease ratio to the increase scenarios 1, 3, and 5. These scenarios represent increasing magnitudes of SOC increases. In scenario 2, SOC is not increased in soil layer 2 at all, and in scenario 4, SOC is increased by 1% (mass) in both layers.

We emphasize that these scenarios are artificial and not intended to represent specific, immediately achievable management interventions, but should rather reflect the long-term possible outcomes of combinations of different management adaptations. While they are informed by the literature review (Sect. 2.1), the scenarios with large and deep SOC increases (MedC\_MedC and

Figure 4. SOC scenarios

vHighC\_MedC) may be harder to achieve in practice. Nevertheless, including such scenarios allows us to explore the potential range of hydrological responses to SOC increases and test the sensitivity of the model to large changes in soil properties.

The SOC values in each scenario are then used to estimate the model input  $\rho_b$  (bulk density), as discussed in Sect. 2.3. Consistent with the rationale outlined in the introduction, we hypothesize that increasing SOC will enhance soil water retention, allowing the soil to buffer hydrologic extremes, reduce low-flow frequency, and modestly attenuate peak flows. mHM considers three land use types: forest, impervious cover and pervious cover, where the latter includes all cropland and meadows. The adaptations to  $\rho_b$  are only applied to pervious areas, which have the highest share in each subcatchment (Fig. 3).

# 3.5 Hydrologic extremes evaluations: Low and peak flow indicators

To assess the impact of the SOC scenarios on low flows, we calculate the Q347 threshold for each sub-catchment. Q347 corresponds to the discharge that is exceeded on 347 days per year (i.e., the 5th percentile) and is commonly used in Switzerland to define low-flow periods and as a threshold for the restriction of irrigation water withdrawal from rivers (Heinz et al., 2025; WPA, 1991).

For peak flows, our analysis is constrained by the daily resolution of simulated discharge, whereas hourly peaks would be more relevant (Bartens et al., 2024). Nevertheless, we estimate changes in discharge associated with two-year return period floods (Q2 events). Q2 thresholds are determined for each sub-catchment using a generalized extreme value model, although the short time series in smaller sub-catchments can be limiting (48 years in the Broye and 26 years in the other subcatchments). Across stations, we also observe a decreasing discharge trend, significant only for the Broye at Payerne, which explains why only a few Q2 events occur during the study period.

#### 4 Results

# 4.1 Calibration and evaluation

Model calibration led to a good fit of simulated to observed streamflow for the Broye and the Flon (KGE = 0.91), with a slightly lower performance for the Arbogne and Petit Glâne (KGE = 0.83 and 0.86, Fig. 5).

**Figure 5.** Observed and simulated discharge for all subcatchments. Except for the Broye, the data for the other stations was not officially validated yet for 2022. NSE= Nash-Sutcliff-Efficiency, pbias = percentage bias.

Seasonal discharge dynamics are not equally well captured across subcatchments (Appendix A). The Broye shows the best fit; low flows are underestimated in the Flon, overestimated and mis-timed in the Arbogne, and mis-timed in the Petit Glâne. Percentage biases for high and low flows (Q95 and Q5) range from -0.1% to 7% and -22% to 43%, respectively, with the best agreement in the Broye (pbias for Q95 = 7%, pbias for Q5 = 9%), likely reflecting the higher quality of observed discharge data there.

In comparison with the observed soil moisture time series (Sect. 3.3), mHM achieved reasonably good performance, except for the lowest soil layer, with KGE values of 0.65, 0.73, and 0.13 (0-30cm, 30-60cm and 60-90cm). The good fit in the two upper layers is noteworthy given that the data were not used for calibration and represent a single grid cell. While soil moisture was generally underestimated (percentage bias 8% to -11%), the model reproduced the temporal variability well (Fig. A1).

# 4.2 Change in soil hydraulic properties

In Fig. 6 (panel a), we show the impact of the SOC increase on several parameters of interest. The points represent all pervious land cover cells, which equals the area the SOC increase is applied to. Saturated hydraulic conductivity ( $K_{\rm sat}$ ) and water content at saturation and wilting point ( $\theta_{Sat}$  and  $\theta_{PWP}$ ) are effective model parameters calculated within mHM.  $\theta_{FC}$  changes with the almost same magnitude as  $\theta_{Sat}$ , which is why we dont show it explicitly in Fig. 6. Bulk density ( $\rho_b$ ) is a model input and PAWC (plant available water capacity,  $\theta_{FC} - \theta_{PWP}$ ) gives an idea if the surplus in retained water could be taken up by plants. The Figure compares the base scenario (no SOC added) with all five SOC increase scenarios.

Adding +0.5%, 1% and 1.5% SOC to the first soil layer led to a decrease in  $\rho_b$  by on average 3.3%, 6.4% and 9.2% (Fig. 6). The decrease in  $\rho_b$  propagates through the model (Fig. 1), leading to an increase in  $\theta_{Sat}$  and  $\theta_{FC}$  of on average 3.2%, 6.2% and 8.8% accordingly. PAWC is increased by 4.9%, 9.3% and up to 13.5%. As described in Sect. 2.3, an increase in  $K_{\text{sat}}$  leads to a decrease in  $\theta_{FC}$ , as parameterized in mHM. Although  $K_{\rm sat}$  was substantially increased, the effect on  $\theta_{FC}$  is negligible. The sensitivity of PAWC to increases in SOC depends on the initial SOC content and soil texture. PAWC increases more strongly in soils with low initial SOC and higher sand content (Appendix A).

# Impact of increasing SOC on grid-scale processes

340 In Fig. 6 (panel b) we visualize how the changes in hydraulic parameters propagate through the process chain in mHM and how states and fluxes are changed accordingly, here representing a snapshot for the simulated net changes. On average, the increase in water retention capacity of the soil leads to an increase in ET and since more water can be retained and evaporated, less water contributes to further states and fluxes downwards.

The overall impacts of the SOC scenarios on the model's fluxes and states at grid-scale are moderate. Fig. 7 shows the results of the base scenario and the SOC scenarios for actual evapotranspiration (ET), soil water content in the first and second soil layer, and subsurface runoff across all permeable land use (grid-) cells throughout the catchment (other states and fluxes are shown in the Supplementary Material). It is important to note, that subsurface runoff in mHM comprises the fluxes fast & slow interflow and baseflow. Soil water content in layer 1 and 2 is with on average 2.9% to 8.1% consistently higher under the SOC scenarios, though spatial variability is large (evident through the large ranges visible in Fig. 7). In absolute terms, soil water content is in winter 3-8 mm and in summer 2-6 mm higher (panel b, Fig. 7). The impact of SOC increase on the boundary fluxes ET and subsurface, is however relatively small. Panel c displays the cumulative absolute differences between each SOC scenario and the base scenario, with relative differences also written in each subplot (Fig. 7). ET shows a net increase of +0.16% to +0.4% over all SOC scenarios, while subsurface runoff slightly decreases (-0.28% to -0.72%). In absolute terms, ET is higher by maximal 0.2-0.6 mm and subsurface runoff is reduced by maximum 1-2 mm. The differences in these key state and fluxes exhibit distinct seasonal patterns. For ET, cumulative differences increase sharply in spring and summer, while little change is observed in fall and winter. In contrast, the cumulative differences in subsurface runoff peak in winter are partly reversed in summer and fall. The difference in soil water content is largest in winter and spring, while it is gradually decreasing in late summer and early fall, before it sharply increases again.

We summarize the average seasonal pattern of all SOC scenarios relative to the base scenario in Fig. 8 and can distinguish four stages, defined in Table 3, which outline the main hydrological responses throughout the year. Overall, we see a consis-360 tent increase in soil moisture across all seasons, moderate increases in ET during spring and summer, and generally reduced subsurface runoff, except in spring when it shows a slight increase (Tab. 3.

#### 4.4 Impact of increasing SOC on discharge and hydrological extremes

Beyond grid-scale subsurface, mHM also simulates routed discharge at the locations of gauging stations. The overall effect of increased SOC on discharge is small. Hydrographs from all scenarios almost entirely overlap and are therefore only shown in

Figure 6. a: Changes in key effective parameters for all pervious landcover cells, which represent the area where SOC was increased in our scenario runs. Please note the differences in scale for each plot. Number in each plot show the mean relative differences for each SOC scenario against the base scenario. b: How changes in SOC and bulk density ( $\rho_b$ ) propagate in the mHM model. The Figure shows a snapshot of net changes in parameters and outputs; actual variables depend on boundary conditions, so seasonal responses may differ. Note that  $\theta_{FC}$  scales linearly with  $\theta_{sat}$  but only weakly with  $K_{sat}$ , leading to an overall increase.  $f_{runoff}$  and I at each timestep are controlled by the current  $\theta$ ; ET affects them only indirectly through its impact on  $\theta$  in preceding timesteps, therefore the dashed arrow here.

Figure 7. All panels show the (weekly) mean and range over all pervious landcover cells, where SOC was increased. a: Timeseries of key fluxes and state for the base and all SOC increase scenarios. Note, that the difference between the scenarios for ET and subsurface runoff is so small that the lines almost completely overlap. b: Absolute difference between each SOC scenario and the base scenario. c: Cumulative sums of the difference between each SOC scenario and the base scenario; the text in each subplot is the mean relative difference over all cells.

Figure 8. Schematic of the annual cycle of average impacts of SOC increase scenarios relative to the base scenario.

380

Table 3. Mean annual hydrological impacts of increased SOC on key fluxes and states relative to base scenario

| Stage/Season             | Soil Moisture       | ET            | Subsurface<br>Runoff | Key Mechanism                                                                                                                                                             |
|--------------------------|---------------------|---------------|----------------------|---------------------------------------------------------------------------------------------------------------------------------------------------------------------------|
| Winter/early spring      | $\uparrow \uparrow$ | No difference | $\downarrow$         | High water retention capacity stores precipitation, minimizing subsurface runoff under SOC scenario.                                                                      |
| Late spring/early summer | 1                   | <b>↑</b>      | <b>↑</b>             | The soil's high initial saturation combined with increasing spring precipitation inputs exceeds the remaining storage capacity, temporarily increasing subsurface runoff. |
| Late summer/fall         | <b>†</b>            | $\uparrow$    | No difference        | A transitional period as soil moisture recovers from the summer peak; no difference in subsurface runoff.                                                                 |
| Late fall/winter         | <b>†</b> †          | No difference | $\downarrow$         | Low ET allows the enhanced retention capacity to<br>maximize SM recovery, reducing subsurface runoff<br>under SOC scenarios.                                              |

Appendix A. Relative differences between the base and SOC scenarios are moderate (positive values indicate higher discharge under SOC; Fig. 9, Panel b), which is consistent with the small changes in subsurface runoff at the grid-cell scale (Sect. 4.3).

All subcatchments display a similar seasonal pattern: the relative difference in discharge decreases mostly in fall and winter, and increases in spring and summer. However, the magnitude and direction of changes differ by subcatchment (Fig. 9 panel b and Fig. 10). Across catchments, relative discharge responses vary in both magnitude and direction. The Flon shows the strongest increases, with values reaching up to +20%, while the Petit Glâne exhibits the largest decreases of up to -18%. For most catchments, relative differences remain within  $\pm 10\%$ . In absolute terms, the Arbogne (mean discharge (mean Q)  $0.73~{\rm m}^3\,{\rm s}^{-1}$ ) shows increases of up to  $0.08~{\rm m}^3\,{\rm s}^{-1}$  and decreases between  $0.10~{\rm and}~0.30~{\rm m}^3\,{\rm s}^{-1}$ . In the Broye (mean Q  $0.34~{\rm m}^3\,{\rm s}^{-1}$ ), discharge can increase by up to  $0.8~{\rm m}^3\,{\rm s}^{-1}$  and decrease by  $1-3.5~{\rm m}^3\,{\rm s}^{-1}$ . For the Flon (mean Q  $0.94~{\rm m}^3\,{\rm s}^{-1}$ ), increases reach up to  $0.05~{\rm m}^3\,{\rm s}^{-1}$ , while decreases range from  $0.05~{\rm to}~0.25~{\rm m}^3\,{\rm s}^{-1}$ . In the Petit Glâne (mean Q  $0.94~{\rm m}^3\,{\rm s}^{-1}$ ), discharge increases are around  $0.1~{\rm m}^3\,{\rm s}^{-1}$ , and decreases range from  $0.3~{\rm to}~0.8~{\rm m}^3\,{\rm s}^{-1}$  (see Appendix A).

The Arbogne resembles the Petit Glâne, and the Broye's response is more attenuated. The share of pervious area per catchment (where SOC is increased) is comparable between the subcatchments: The Broye and Arbogne have 61% and 62%, Flon and Petit Glâne slightly higher shares with 72% and 74% (Fig. 3). Thus, the described differences between stations rather arise from climatic variations than differences in land use.

**Figure 9.** Relative difference timeseries of SOC to base scenario. positive values = more discharge under SOC increase. Gray vertical lines = days where the low flow threshold for each subcatchment is reached in the base scenario, red vertical lines = days where Q2 threshold is reached.

**Figure 10.** Spatial patterns of precipitation, temperature and discharge for one intermediate SOC scenario (MedC\_lowC). Difference in discharge = SOC-scenario - base scenario. More details and monthly maps in Appendix A

The Flon chatchment has a higher average elevation, with lower temperatures, more precipitation and therefore increased discharge (Fig. 10). The Petit Glâne and Arbogne lie lower and receive less precipitation and show therefore also less discharge. The Broye subcatchment spans a wider elevational and climatic gradient, thus slightly averaging out the effects.

Peak flows are, in general, reduced under the SOC scenarios, although the effect is small (Fig. 9). Floods with a 2-year return period occurred in winter 2017/2018 and summer 2021 (Q2 events, red vertical lines in Fig. 9). Discharge during these events is slightly decreased under the SOC scenarios in 2017/2018, but the impact in 2021 is negligible. For instance, the peak flow in the Broye in 2018 reached 77.17 m<sup>3</sup>s<sup>-1</sup> and was reduced by 0.2–0.5 m<sup>3</sup>s<sup>-1</sup> under the SOC scenarios, while in 2021 a peak of 70.29 m<sup>3</sup>s<sup>-1</sup> was reduced by 0.3–1 m<sup>3</sup>s<sup>-1</sup> (see Appendix A).

A relevant indicator for low flows is the Q347 threshold, an indicator that cantonal authorities use to determine bans on irrigation water withdrawal from rivers to fulfill the minimum environmental flow requirements. In Fig. 9, days where the observed discharge fell below this threshold are marked in gray as low-flow days. Although the influence is minor, discharge is slightly increased under the SOC scenarios before and sometimes during observed low-flow periods. This leads to fewer days falling below the Q347—typically 1–6 days depending on scenario, year, and subcatchment (in the Broye, for example, 1–4 days less). However, in the Arbogne in 2016 and 2019, as well as in the Petit Glâne in 2019, low-flow periods coincided with reduced discharge under the SOC scenarios, resulting in more low-flow days (a surplus of 1–14 days in the Arbogne and up to 5 days in the Petit Glâne, Fig. 11).

**Figure 11.** Timeseries of relative difference between base and SOC scenarios in the annual number of days with discharge below Q347 (low flow threshold).

# 4.5 Scenario sensitivity

Since the overall small impact of SOC increase on ET was first surprising, we wanted to investigate the responses of the individual soil layers. Here we found, that although the soil water content in the first two layers was consistently higher under the SOC scenarios, ET from soil layers 1 is reduced, while it is increased from soil layer 2 and 3, leading to an overall small net increase. The reason for this is explained and discussed in Sect. 5.2.3.

The SOC scenarios represent possible outcomes of combinations of management adaptations. Their impact on the model output fluxes ET and total grid-scale runoff increases almost linearly with increasing SOC content, as visible by comparing scenarios LowC\_vLowC, MedC\_LowC and vHighC\_MedC in Fig. 12.

In total, the largest SOC additions occur in scenario vHighC\_MedC (+1.5% in the first layer and +0.6% in the second layer), whereas in scenario MedC\_MedC, SOC is added evenly across both layers (+1% in each). Interestingly, the effects on soil moisture are often largest under MedC\_MedC, despite its slightly lower total SOC increase (Fig. 7). This suggests that increasing SOC in the subsoil can be particularly beneficial, as it slows soil moisture depletion during late summer and fall.

Figure 12. Magnitude of change in selected states and fluxes for SOC scenarios relative to the base scenario.

Differences between vHighC\_MedC and MedC\_MedC are generally minor: in most catchments, the two scenarios produce nearly identical reductions in low-flow days compared to the base scenario. However, in the Arbogne in 2018, low-flow days are reduced by one day only under MedC\_MedC, and in the Flon in 2022, the reduction under scenario MedC\_MedC is two days—one day more than under vHighC MedC.

For the catchment-wide, annual effect, the distribution of SOC in the two soil layers does not make a difference. Only at the seasonal scale, a distribution into deeper layers might lead to a delay of drought-induced transpiration reduction (as was observed in Turek et al. 2023; Heinz et al. 2025).

# 5 Discussion

# 5.1 Applicability of the study framework

#### 5.1.1 Model performance

In this study, we used a fixed parameter set that was calibrated to perform well across all four subcatchments, effectively representing observed discharge dynamics. Model or parameter uncertainty was not systematically explored, but testing six alternative parameter sets showed that the direction of simulated changes is robust, while the magnitude varies (Sect. 5.2.2).

For streamflow, the calibrated mHM model performs very well for the Broye (Payerne) subcatchment (KGE = 0.91, NSE = 0.86), outperforming previous applications of conceptual models (SWAT, Zarrineh et al. 2018; PREVAH, Muelchi et al. 2021b) and a physics-based model (Alpine3D, Lehning et al. 2006). Despite relatively short calibration and evaluation periods (four and three years), these performance values are high (see Supplementary Material for a more detailed comparison), underlining the model's ability to reproduce observed discharge dynamics. Seasonal low-flow regimes are fairly well reproduced for the Broye and Petit Glâne subcatchments, while the frequency of low flows is underestimated for the Flon and overestimated for the Arbogne. The differences in performance can be traced to biases in the precipitation input fields. Such biases were already reported in earlier studies using the same precipitation data product (Muelchi et al., 2021b; Brunner et al., 2019). While our adjustments to the precipitation input substantially reduced these biases, they were not fully eliminated. A more systematic bias correction would be required. Still, this adaptation was essential to reliably simulate soil moisture dynamics. Here, we were able to reproduce observed soil moisture time series with relatively good performance (Appendix A).

# 5.1.2 Plausibility of represented changes in soil hydraulic properties

The plausibility of our simulation results depends on how reliably SOC-driven changes in soil hydraulic properties are represented, which can only be discussed against literature reported estimates.

The non-linear pedotransfer function (PTF) used to calculate  $\rho_b$  from SOC captures the stronger sensitivity of  $\theta_{FC}$  and  $\theta_{Sat}$  at low initial SOC, consistent with large soil database analyses (Rawls et al., 2004; Hudson, 1994; Minasny and McBratney, 2018). Simulated reductions in  $\rho_b$  (Fig. 6) align with reported ranges (Table 1). In our study,  $\theta_{FC}$  and  $\theta_{Sat}$  increase with SOC, while  $\theta_{PWP}$  increases less, raising plant available water capacity (PAWC), in line with previous findings (Rawls et al., 2003, 2004; Libohova et al., 2018; Lal, 2020; Abdallah et al., 2021). Exact incremental changes along the retention curve remain however uncertain and are soil-specific (Lal, 2020). Reported PAWC increases vary widely: 1.5–7% for +0.6% SOC (Libohova et al., 2018; Archer and David, 2005), 1.16% for +1% SOC (Minasny and McBratney, 2018), up to 50% for +1.5% SOC (Libohova et al., 2018), and 4–45% for management-related SOC gains (7–220% (Haruna et al., 2020; Hao et al., 2023; Blanco-Canqui et al., 2023)). Our simulated average increase of 9.3% for +1% SOC (+1% (mass) increase  $\approx$  35–60% relative increase) lies within these ranges but toward the upper end. Changes in  $\theta_{Sat}$  or  $\theta_{FC}$  are rarely quantified, but Shi et al. (2016) reported +8–10% in a silt loam, comparable to our +6.2% average. Changes in  $\theta_{PWP}$  are unfortunately rarely reported, meaning that we cannot explicitly assess the plausibility of our simulated average increase in  $\theta_{PWP}$  by 2.1%. The SOC effect on water retention is texture dependent, with greater PAWC increases in coarser soils (Lal, 2020; Libohova et al., 2018) and at low initial SOC (Rawls et al., 2004), patterns consistent with our results (Appendix A).

The simulated change of  $K_{\rm sat}$  aligns with observations from Haruna et al. (2020); Hao et al. (2023); Blanco-Canqui et al. (2023) for similar soils, is higher than those reported by Veettil et al. (2024); Rawls et al. (2004), but comparable to Kojima et al. (2018); Bormann et al. (2007), who also used PTF estimates rather than observations, which carry their own assumptions and uncertainties. Given high variability in pedo-climatic conditions, management practices, PTF selection, and soil texture, the plausibility of simulated hydraulic changes can only be assessed generally: overall trends and magnitudes are plausible,

475

but uncertainty remains, especially for  $K_{\text{sat}}$  and  $\theta_{PWP}$ . A potential way forward would be to incorporate additional soil probe measurements and observationally derived hydraulic parameters, such as  $K_{\text{sat}}$ , to further constrain and validate the model.

#### 5.2 Simulated impact of management adaptations

# 5.2.1 SOC enhances ET and reduces subsurface runoff at the grid-scale

Although the SOC scenarios considered here are based on simplified and rather ambitious assumptions (i.e., uniform applicability of management across all permeable cells), their overall impact on simulated hydrological processes is small. This can be attributed to the moderate changes in hydraulic properties following SOC application, which fall within observed ranges and can therefore be considered plausible (Sect. 5.1.2). Under the strongest SOC scenario (vHighC\_MedC), θ increases by 8.8%; under the weakest (LowC\_vLowC) by 3.2% (Fig. 7). This increased retention capacity extends the water content in the top two soil layers, reducing percolation and temporarily enabling higher ET during summer when evaporative demand peaks and when ET would otherwise be water-limited. Over 2016–2022, this results in a net ET increase of +0.16–0.4% (8–18 mm depending on scenario, Fig. 7).

Direct comparison with experiments is difficult since ET is rarely measured; PAWC is often used as a proxy due to its influence on transpiration and yield (Feifel et al., 2023). Skadell et al. (2025) analyzed SOC effects on soil hydraulic properties across European sites and found that, although overall impacts on water retention were limited, SOC increased PAWC and slightly delayed plant drought stress. This implies a modest rise in transpiration, but it was not measured directly. In a lysimeter experiment Ghanem et al. (2022) showed that biochar application to sandy soil reduced bulk density, increased porosity, and ultimately enhanced ET.

Plot-scale simulations with Richards-equation-based models show that +1% (mass) SOC can increase transpiration by up to 9%, while soil evaporation may decrease due to higher crop cover (Heinz et al., 2025). Turek et al. (2023) reported similar increases (+15 mm year<sup>-1</sup> for +1% SOC down to 65 cm). Using Hydrus-1D (Šimůnek et al., 2013), Feifel et al. (2023) found that evaporation always increased, whereas transpiration rose only when SOC increased below 30 cm depth, with stronger effects in finer soils and drought years. Deep drainage and recharge consistently declined, in line with our simulated percolation and recharge decreases.

Unlike these plot-scale studies, mHM represents ET as a single bulk flux after canopy interception, without separating evaporation and transpiration or explicitly limiting root water uptake. Consequently, direct comparison with plot-scale studies is limited. We argue, the simulated ET increase in this study is likely dominated by transpiration, which the increase in PAWC suggests (Fig. 6). During high summer evaporative demand, the increased PAWC and  $\theta$  translate directly into higher ET, slightly reducing subsurface runoff (-0.3% to -0.78%, 10–22 mm over 7 years).

The actual impact of SOC on ET strongly depends on management: cover crops, mulching, or residue retention can suppress soil evaporation (Abdallah et al., 2021). The effect of topsoil changes is two-fold: soil cover changes like mulching can i) modify the re-evaporation of (soil-)intercepted water and ii) impact soil evaporation, i.e. of water that infiltrated into the topsoil (Ramos et al., 2024). The first could, in principle, be represented through modified interception parameterizations; the

second remains largely absent from current catchment-scale models and represents a key direction for future work—explicitly distinguishing soil evaporation and transpiration across temporal and vertical scales.

In summary, modest SOC increases slightly enhance summer ET, likely via transpiration, and marginally reduce subsurface runoff. These effects align with field-scale findings and modeling results, particularly where SOC increases extend deeper into the soil, underscoring the importance of considering both depth and method of SOC application in agricultural practice. The simulated reduction of deeper drainage, and thus recharge, in our and other modeling studies highlights a potential trade-off between enhancing SOC for agricultural benefits and sustaining hydrologic processes critical for water management, especially under changing climate conditions.

Although it is beyond the scope of this study, we acknowledge that increasing SOC also affects soil biogeochemical cycles, particularly when nutrient balances change. For example, in poorly-drained soils, increasing SOC without adjusting nitrogen inputs can enhance denitrification and lead to elevated emissions of  $N_2O$ , a potent greenhouse gas, thus representing a trade-off worth noting (Jäger et al., 2011).

## 5.2.2 Catchment-scale implications of SOC-induced changes in discharge

SOC-related impacts on discharge are seasonally dependent. In spring, increased rainfall combined with high soil moisture ( $\theta$ ) levels from winter, pushing more water into percolation and discharge just before the low-flow period. During the low-flow period itself, higher ET reduces percolation, so discharge increases are limited (1–5%) or may even turn into decreases. These modest discharge increases can reduce days below the Q347 threshold by 1–6 compared to the base scenario, potentially easing irrigation constraints (Heinz et al., 2025). This holds for the Flon and Broye, but in the Arbogne (2016, 2019) and Petit Glâne (2017, 2019), low-flow days mostly increase. These two subcatchments are the lowest, thus warmer and also drier than the others. When low subsurface runoff coincides with high ET, the SOC scenarios further enhance water retention and ET, which can exacerbate discharge reductions and increase low-flow days, potentially increasing the likelihood of irrigation constraints.

Seasonal discharge dynamics are best captured by the model in the Broye; smaller subcatchments show biased or mis-timed low flows, likely reflecting input data limitations. Changes in low-flow days, derived from the 95th percentile of discharge, are sensitive to model optimization and thus less robust. While overall discharge fits are nearly identical across six optimization runs, variation at the distribution tails is observed and expected (Appendix A). The selected optimization run reflects the overall pattern observed over several optimization runs: reductions in low-flow days are consistent in the Broye and Flon, whereas the pattern for the Arbogne and Petit Glâne are more variable. However, the number of low flow days most often increases in 2016, 2017 and 2019. This suggests that SOC is likely to reduce low-flow days in larger, or cooler and wetter catchments, but impacts in smaller, warmer, and drier catchments are highly variable and climate-dependent, often even increasing the number of low flows. Small catchments, with typically low storage and fast hydrological response, are highly sensitive to minor changes in precipitation and temperature (Thomas et al., 2011). Thus, even modest SOC-induced reductions in discharge can push flows below ecological or minimum thresholds, making these trade-offs especially relevant in smaller catchments under future climate changes.

Observed discharge in the Broye peaks between December and March, but SOC-related reductions occur mainly in late autumn and winter, not necessarily the most critical period (Fig. 9). Q2 events in winter 2017/2018 and summer 2021 show only moderate reductions (Appendix A). Daily simulation resolution prevents precise quantification of peak reductions.

Evidence on agricultural management effects on peak and low flows is limited, with most studies focusing on major land-use changes or structural interventions. As a result, direct validation of our findings is challenging, and comparisons must be made cautiously. In a modeling experiment on land use changes, Moussa et al. (2002) found that high  $K_{\text{sat}}$  reduces mean discharge and flood peaks, consistent with our findings, although we only found a very limited effect of increased  $K_{\text{sat}}$  on  $\theta_{\text{FC}}$ . Antolini et al. (2019) simulated the impact of cover crops and reduced tillage and found a moderate reduction in high-frequency flood peaks, also in line with our results. Similarly, Fatichi et al. (2014) used Tethys-Chloris to show that strong soil compaction at the plot scale (represented by -95%  $K_{\text{sat}}$ ) can increase discharge peaks by 50–80% at the catchment scale.

Our SOC scenarios, though spanning different magnitudes, are optimistic, assuming SOC increases across all pervious cells. This assumption is constrained, as pervious cover aggregates cropland and meadows, which at the moment cannot be separated in the model. Upstream, where meadow cover increases, substantial SOC gains below 30 cm are less likely. Even under these assumptions, the catchment-scale effects are very modest, which is in line with Fatichi et al. (2014), suggesting detectable impacts of management require either strong interventions or long observation periods.

In recent years, a broader debate around nature-based solutions and soil and water conservation measures has emerged. Several international and European initiatives aim to enhance soil carbon sequestration, soil health, and water retention through nature-based and conservation practices. The 4 per mille Initiative (https://sdgs.un.org/partnerships/4-1000-initiative-and-its-implementation) promotes the increase of global SOC stocks by 0.4% per year to offset CO<sub>2</sub> emissions from fossil fuels (Minasny et al., 2017). The EU project NBsoil (https://nbsoil.eu/) focuses on nature-based soil management to enhance soil ecosystem services. The EJP Soil project SoilX (https://projects.au.dk/ejpsoil/soil-research/eom4soil/into-dialogue/soilx) develops strategies to improve soil carbon, soil health, and water retention. The OPTAIN project (https://www.optain.eu/) promotes small water retention measures and nutrient management in agricultural catchments. While increasing SOC can enhance water retention, slightly reduce flood peaks, and decrease low-flow frequency, the catchment scale benefits remain modest even under large SOC increases. Moreover, our results indicate that in smaller, drier agricultural catchments, SOC-enhancing measures may involve trade-offs, such as reduced groundwater recharge or streamflow, reducing downstream water availability, which should be considered when designing management strategies.

# 5.2.3 Root distribution dependency on $\theta_{FC}$ in mHM

As noted in Sect. 4.5, the SOC-induced increase in  $\theta$  only led to a very small net increase in ET, since ET from the top soil layer actually decreased, which was unexpected. This response stems from the mHM adaptation by Demirel et al. (2018), which links root distribution to field capacity ( $\theta_{FC}$ ): higher  $\theta_{FC}$  shifts root fractions (R) downward, reducing the weight of the top layer and increasing that of the lower ones. This relationship was derived from observations in the region where the scheme was developed, where sandy soils with low  $\theta_{FC}$  concentrated roots near the surface, while clay-rich soils with high

585

 $\theta_{FC}$  showed deeper rooting (Demirel et al., 2018). Yet as noted by Demirel et al. (2018), such a pattern is not necessarily globally valid.

If we recall the formulation of the soil moisture stress function f (Sect. 2.3), which linearly scales PET to ET, we saw that f depends on the root fraction R and the normalized soil water content  $\overline{\theta}$  (calculated as:  $\overline{\theta} = (\theta - \theta_{\rm pwp})/(\theta_{\rm sat} - \theta_{\rm pwp})$ ).

If mean soil water content  $\bar{\theta}$  increases or decreases depends on the SOC-induced increases in  $\theta_{pwp}$  and  $\theta_{sat}$  that are texture dependent, but also on the daily varying  $\theta$ , which depends on incoming precipitation, so seasonality. Only if  $\theta$  increases sufficiently,  $\bar{\theta}$  would increase and by that also f and hence ET. This mechanism applies in principle to both upper soil layers, but in the top layer R decreases as  $\theta_{FC}$  increases. Consequently, even though  $\bar{\theta}$  tends to increase, the overall stress factor f (and thus ET) decreases in most cases. Only under very wet conditions, high soil water content  $\theta$  may offset the reduction in root fraction and ET can still increase. In deeper layers, the opposite holds: the higher root fraction allows f to increase, so ET is increased when additional water infiltrates from above. Physiologically, this is unexpected, as plants usually allocate roots cost-efficiently to shallow layers where water and nutrients are accessible, though they may extend them deeper under drought (Jarvis, 1989; Fry et al., 2018; Jackson et al., 1996; Maan et al., 2023). More broadly, root allocation depends on cultivar and growth stage (Tajima, 2021), and such dynamics are difficult to generalize at the catchment scale. Nevertheless, as discussed in Sect. 5.2.2, the overall pattern of SOC-induced changes remains robust. Future work could test how increased  $\theta_{FC}$  affects root depth allocation and evapotranspiration dynamics under local climatic and edaphic conditions in our case study region.

Note that in Fig. 12, soil water content in the top layer differs slightly among the three SOC scenarios MedC\_MedC, MedC\_LowC and MedC\_ZeroC, even though the SOC increase in this layer is identical. This results from the top-down calculation of the root fraction R per layer and subsequent re-normalization, which ensures that R sums to one across all layers. Therefore, differences in SOC in the lower soil layers can indirectly impact soil water content and ET in the top layer.

# 5.2.4 Role of SOC increase magnitude and depth in modulating hydrological responses

The SOC scenarios affect ET and subsurface runoff almost linearly with increasing SOC. Seasonal differences emerge when SOC is distributed into deeper layers: Scenario MedC\_MedC (+1% SOC in both soil layers) exhibits the highest increase in θ over the winter and spring, despite vHighC\_MedC adding more SOC in total (Fig. 7). Adding SOC to deeper layers delays overall soil moisture depletion and can thus reduce drought impacts, which was also concluded in the modeling studies of Turek et al. (2023) and Feifel et al. (2023). In our model simulations on catchment-annual scales, however, the vertical SOC distribution plays little role and achieving a significant increase in SOC in deeper layers is more difficult, as most (agricultural) adaptation measures would primarily lead to SOC increases near the surface (Bai et al., 2019).

#### 5.3 Model suitability and structural limitations for representing SOC-induced changes

The mHM model is well suited for impact studies like this due to its open-source nature, active user community, and flexible structure, which allows individual adjustments, such as in the estimation of soil hydraulic properties (Livneh et al., 2015).

As with the case of any modeling scheme, some simplifications and limitations remain. First, in mHM only three land-use classes are distinguished, which may be sufficient at large scales but limits the representation of heterogeneous agricultural

landscapes. In our study region, pervious land cover aggregates cropland and meadows, which can differ in management and water-use processes. Introducing additional land-use classes and distinguishing different crop functional types with varying root profiles would improve model realism.

Differentiating winter crops and spring crops could be important too, since they have distinct patterns of water uptake and thus may influence recharge and also low flow dynamics differently.

Regarding ET, mHM separates canopy interception but combines soil evaporation and transpiration into a single flux, common in many hydrological models (Samaniego et al., 2010). While net ET is likely captured correctly, partitioning between productive (transpiration) and unproductive (soil evaporation and interception) fluxes, as well as their temporal dynamics, may differ from reality. Finally, the root distribution, which varies with  $\theta_{FC}$ , is more dynamic than standard static profiles, but could be improved by incorporating dynamic crop/root growth and reassessing the negative relationship between  $\theta_{FC}$  and root density under the climatic and edaphic conditions of our study region. Overall, while these limitations can affect processes and fluxes at the plot scale, their impact at the catchment scale is uncertain, particularly given the lack of direct observations for root distribution or ET partitioning. Compared to fully physics-based models such as WaSiM-ETH (Schulla, 1997), which often require extensive parameter adjustment and high computational effort, mHM offers a practical balance between spatially explicit process representation and computational efficiency (Samaniego et al., 2010; Kumar et al., 2013; Samaniego et al., 2017). Fully physics-based models are in practice never "fully" mechanistic, and for our purpose they would not provide additional advantages in representing SOC-related management effects. Their higher data and computational demands would mainly add complexity without improving the core processes relevant to this study.

#### 605 6 Conclusion

We explored how increasing soil organic carbon (SOC) through agricultural management could alter catchment-scale hydrology, using the Broye catchment in Western Switzerland as a representative case study. By representing SOC-driven changes in soil hydraulic properties in a large-scale hydrological model (mHM), we traced how increased water retention could affect evapotranspiration, subsurface runoff, and streamflow extremes. We applied five SOC scenarios varying in depth and magnitude to explore process sensitivity. While the direction and timing of SOC effects are credible, their magnitude remains uncertain due to limitations in pedotransfer functions and parameterization. At the plot scale, the increase in SOC increased soil water content (2.99–8.13%), slightly increased evapotranspiration (0.15–0.4%), and marginally reduced subsurface runoff (0.28–0.72%), depending on the applied SOC scenario. At the catchment scale, effects were highly context-dependent: SOC-driven improvements in soil water retention tended to support higher evapotranspiration but reduced groundwater recharge and discharge, a clear trade-off. These shifts occasionally intensified low flows in warmer and drier subcatchments (Arbogne, Petit Glâne), while they could temporarily alleviate them in cooler and wetter areas (Broye, Flon), especially under deeper SOC increases.

Our key takeaways are:

Even optimistic and substantial increases of SOC, and thus changes in hydraulic properties, lead to relatively modest
 impacts at the catchment scale.

The hydrological effects of SOC management depend strongly on local hydro-climatic conditions: the intended increase
in plant-available water can reduce critical low-flow periods. However, it can also lead to unwanted ET increases and
slightly reduce summer discharge.

Future work should focus on capturing vegetation and transpiration dynamics more accurately, including the interplay
of crops with different growing seasons (winter vs. spring crops), to improve model realism.

Overall, our analysis emphasizes the need for a better understanding of the trade-offs and balances between agricultural practices aimed at increasing soil organic carbon (SOC) – including initiatives such as the 4 per mille and other soil carbon sequestration efforts – and their resulting impacts on catchment hydrological processes, ranging from soil moisture dynamics to groundwater recharge and hydrologic extremes.

Code availability. Scripts to pre- and postprocess and visualize mHM input and output data: 10.5281/zenodo.17515165. The mHM source code is available on the developers github: https://github.com/mhm-ufz/mhm.

*Data availability.* The adapted precipitation timeseries (explained in further detail in Sect. 3.2 and in the Supplementary Material) is available here: 10.5281/zenodo.17243146.

Author contributions. MH, BS, AH, PH and RK contributed to conceptualization; MH, BS, AH, PH and RK to methodology; MH, SL PH and RK to software; MH to validation; MH to formal analysis; MH to investigation; PH, BS and RK to resources; MH, PH and SL to data curation; MH, PH and SL to writing – original draft; MH, BS, AH PH and RK to writing – review & editing; MH to visualization; AH, BS and RK to supervision.

Competing interests. One of the authors (RK) is editor of this journal.

Acknowledgements. We thank Christoph Raible and Natalie Ceperley for stimulating discussions on the manuscript. Some portions of the text were drafted with assistance from OpenAI's ChatGPT (GPT-5). No sentences were taken literally from generated text. The author reviewed, edited, and takes full responsibility for the content. The software BioRender was used in the creation of figures.

# Appendix: A1

**Table A1.** Mean annual water balance components (2016–2022) for four subcatchments under three model runs. Where i) is the run with the default parameter set and the default RhiresD precipitation input data, ii) is the default parameter set but with the adjusted precipitation input data (RhiresD+) and iii) is the optimized parameter set and precipitation input data.  $Q_{\rm obs}$  and  $Q_{\rm sim}$  denote the observed and simulated discharge, P is precipitation and ET is simulated evapotranspiration. Values in mm yr<sup>-1</sup>.

| Run                         | Subcatchment       | $Q_{ m obs}$ | $Q_{\rm sim}$ | P    | ET  |
|-----------------------------|--------------------|--------------|---------------|------|-----|
|                             | Broye (Payerne)    | 504          | 509           | 1120 | 586 |
| (') D. C. Iv. DI.' D        | Petit Glâne (Cugy) | 298          | 385           | 917  | 566 |
| (i) Default + RhiresD       | Arbogne (Avenches) | 244          | 370           | 867  | 577 |
|                             | Flon Aval (Oron)   | 527          | 654           | 1285 | 615 |
|                             | Broye (Payerne)    | 504          | 509           | 1120 | 586 |
| (ii) Default + RhiresD+     | Petit Glâne (Cugy) | 298          | 373           | 917  | 545 |
|                             | Arbogne (Avenches) | 244          | 285           | 867  | 554 |
|                             | Flon Aval (Oron)   | 527          | 654           | 1285 | 615 |
|                             | Broye (Payerne)    | 504          | 490           | 1120 | 612 |
| ("') C 1"                   | Petit Glâne (Cugy) | 298          | 324           | 917  | 598 |
| (iii) Calibrated + RhiresD+ | Arbogne (Avenches) | 244          | 260           | 867  | 589 |
|                             | Flon Aval (Oron)   | 527          | 644           | 1285 | 635 |

# Appendix: A2

The following equations are used to estimate van Genuchten parameters and other key soil hydraulic properties (assuming soil texture given in fractions [0-1]):

$$\theta_{\text{sat}} = P_{\text{constant}} + P_{\text{clay}} \cdot T_{\text{clay}} + P_{\text{BD}} \cdot \rho_b \tag{A1}$$

$$n = C_{\text{vG1}} - C_{\text{vG2}} \cdot T_{\text{sand}}{}^{C_{\text{vG3}}} + C_{\text{vG4}} \cdot T_{\text{clay}}{}^{C_{\text{vG5}}}$$
(A2)

$$m = 1 - \frac{1}{n} \tag{A3}$$

$$\alpha = \exp\left(C_{\text{vG6}} + C_{\text{vG7}} \cdot T_{\text{sand}} + C_{\text{vG8}} \cdot T_{\text{clay}} - C_{\text{vG9}} \cdot \rho_b\right) \tag{A4}$$

650 
$$\theta_{FC} = \theta_{\text{sat}} \cdot \exp\left(C_{FC1} \cdot \left(C_{FC2} + \log_{10}(K_{\text{sat}})\right) \cdot \log(n)\right) \tag{A5}$$

$$\theta_{\text{PWP}} = \frac{\theta_{\text{sat}}}{\exp\left(m \cdot \log\left(C_{\text{PWPc}} + \exp\left(n \cdot \log\left(\alpha \cdot C_{\text{PWPh}}\right)\right)\right)\right)} \tag{A6}$$

All constant and parameter values are listed in Tables A2 and A3.

Equation for  $\theta_{FC}$  depends on paper by (Twarakavi et al., 2009), that calculates  $\theta_{FC}$  in dependence of  $K_{sat}$ , decreasing  $\theta_{FC}$  with increasing  $K_{sat}$ .

Changes in SOC and bulk density ( $\rho_b$ ) would propagate to other soil hydraulic parameters, as evident in equations 7 to A6. These would affect the estimation of  $\theta$ , which initialized to = 0.25 and then updated at each timestep via:

$$I = \begin{cases} P_{\text{effective}}, & \text{if } \theta > \theta_{\text{sat}} \\ P_{\text{effective}} + (\theta - \theta_{\text{sat}}), & \text{if } \theta + x_{\text{tmp}} > \theta_{\text{sat}} \\ P_{\text{effective}} - x_{\text{tmp}}, & \text{otherwise} \end{cases}$$
(A7)

$$\theta_{\text{new}} = \begin{cases} \theta_{\text{sat}}, & \text{if } \theta + x_{\text{tmp}} > \theta_{\text{sat}} \\ \theta + x_{\text{tmp}}, & \text{otherwise} \end{cases}$$
(A8)

$$f_{\text{runoff}} = \begin{cases} \exp(e_{\text{soil\_moisture}} \log(\frac{\theta}{\theta_{\text{sat}}})), & \theta > 0 \\ 0, & \text{otherwise} \end{cases}$$
(A9)

$$660 \quad x_{tmp} = P_{effective} \left( 1 - f_{runoff} \right) \tag{A10}$$

Where  $P_{\text{effective}}$  is either incoming precipitation or Infiltration (I) from the above soil layer. The change in  $\theta$  then again propagates to the root zone soil moisture storage ( $X_3$  in Figure 1):

$$X_{3} = I^{(k-1)} - \mathrm{ET}^{k} - \left(I^{(k-1)} - \left(I^{(k-1)} \left(1 - \mathrm{f}_{\mathrm{runoff}}\right)\right)\right) \tag{A11}$$

Where k is the soil layer and I is the Infiltration coming from the layer above and  $e_{\text{soil}}$  moisture is being calibrated.

# 665 Appendix: A3

Not only do the changes in soil hydraulic properties affect soil moisture, but the soil moisture also governs how much water can evapotranspire from each layer:

$$\overline{\theta_{FC}} = \frac{\theta_{FC} - \theta_{min}}{\theta_{max} - \theta_{min}} \tag{A12}$$

where  $\theta_{\text{max}}$  is  $\theta_{\text{global}} + \theta_{\text{min}}$ , and:

$$R_{\text{CoeffFC}} = \overline{\theta_{\text{FC}}} \cdot f_{\text{clay}} + (1 - \overline{\theta_{\text{FC}}}) \cdot f_{\text{sand}}$$
(A13)

# Appendix: A4

Table A2. Constants used in mHM. Column "Constant" refers to the name used in this study.

| Constant            | Value     | Description                        | Name in mHM       |
|---------------------|-----------|------------------------------------|-------------------|
| $C_{\rm FC1}$       | -0.60     | Constant in PTF for $\theta_{FC}$  | field_cap_c1      |
| $C_{\mathrm{FC2}}$  | 2.00      | Constant in PTF for $\theta_{FC}$  | field_cap_c2      |
| $C_{ m PWPc}$       | 1         | Constant in PTF for $\theta_{PWP}$ | PWP_c             |
| $C_{\mathrm{PWPh}}$ | 15000     | Constant in PTF for $\theta_{PWP}$ | PWP_matPot_ThetaR |
| $C_{ m vG1}$        | 1.392     | Constant in PTF for $n$            | vGenuchtenN_c1    |
| $C_{ m vG2}$        | 0.418     | Constant in PTF for $n$            | vGenuchtenN_c2    |
| $C_{ m vG3}$        | -0.024    | Constant in PTF for $n$            | vGenuchtenN_c3    |
| $C_{ m vG4}$        | 1.212     | Constant in PTF for $n$            | vGenuchtenN_c4    |
| $C_{ m vG5}$        | -0.704    | Constant in PTF for $n$            | vGenuchtenN_c5    |
| $C_{ m vG6}$        | -0.648    | Constant in PTF for $\alpha$       | vGenuchtenN_c6    |
| $C_{ m vG7}$        | 0.023     | Constant in PTF for $\alpha$       | vGenuchtenN_c7    |
| $C_{ m vG8}$        | 0.044     | Constant in PTF for $\alpha$       | vGenuchtenN_c8    |
| $C_{ m vG9}$        | 3.168     | Constant in PTF for $\alpha$       | vGenuchtenN_c9    |
| $C_{Ksat1}$         | 1.1574e-7 | Constant in PTF for $K_{\rm sat}$  |                   |
| $C_{Ksat2}$         | 20.62     | Constant in PTF for $K_{\rm sat}$  |                   |
| $C_{Ksat3}$         | 0.96      | Constant in PTF for $K_{\rm sat}$  |                   |
| $C_{Ksat4}$         | 0.66      | Constant in PTF for $K_{\rm sat}$  |                   |
| $C_{Ksat5}$         | 0.46      | Constant in PTF for $K_{\rm sat}$  |                   |
| $C_{ m Ksat6}$      | 8.43      | Constant in PTF for $K_{\rm sat}$  |                   |

Appendix: A5

Appendix: A6

According to Ruehlmann and Körschens (2009) and Robinson et al. (2022), the rate of change in bulk density ( $\rho_b$ ) following an increase in SOC should increase with increased initial  $\rho_b$ . This is not the case, however, in the current representation of how SOC can affect  $\rho_b$  in mHM. Currently, input to the model should be mineral  $\rho_b$  (Dbm), which is then adjusted to total  $\rho_b$  (Db):

$$Db = \frac{100}{\left(\frac{pOM}{\text{BulkDens\_OrgMatter}}\right) + \left(\frac{100 - pOM}{\text{Dbm}}\right)}$$
(A14)

Where pOM is a calibration parameter adjusting the percentage of organic matter in the soil (statically, over all cells), and  $BulkDens\_OrgMatter$  is the  $\rho_b$  of organic matter. Since we want to be able to represent different scenarios of SOC increases, we also want this to be reflected in the  $\rho_b$ , as and increase in SOC has been observed to decrease  $\rho_b$  (Rawls et al., 2004; Chalise et al., 2019; Haruna et al., 2020). To overcome the limitation of adjusting the pOM parameter statically for all cells, we mute this part in the model and instead use already adjusted  $\rho_b$  values as input. To this end, we estimate  $\rho_b$  from initial and perturbed distributed SOC input data via the PTF by Manrique and Jones (1991), adopted by De Vos et al. (2005):

$$\rho_b = 1.660 - 0.318\sqrt{\text{SOC}} \tag{A15}$$

# 685 Appendix: A7

680

**Figure A1.** Observed and simulated (mHM) timeseries of volumetric water content at the SwissSMEX grassland site near Payerne. Simulations represent three soil layers, while observations are point-scale: layer 1 (5, 10, 15 cm, integrated), layer 2 (50 cm), and layer 3 (80 cm).

**Figure A2.** Discharge for all subcatchments and SOC scenarios. Gray vertical lines = days where the low flow threshold for each subcatchment is reached in the base scenario, red vertical lines = days where Q2 threshold is reached.

Figure A3. Absolute difference in discharge between the base and each SOC increase scenarios.

Figure A4. Observed and simulated low flow days for all subcatchments.

**Table A1.** Metrics for peak flow and low flow fit: Q95 and Q5 denote high- and low-flow percentiles; Peak\_bias and Low\_bias are percent biases (%).

| Station     | KGE  | Q95_obs | Q95_sim | Q5_obs | Q5_sim | Peak_bias | Low_bias |
|-------------|------|---------|---------|--------|--------|-----------|----------|
| Petit Glâne | 0.86 | 2.63    | 2.66    | 0.17   | 0.20   | 1.24      | 18.09    |
| Arbogne     | 0.83 | 1.77    | 1.76    | 0.20   | 0.15   | -0.08     | -22.51   |
| Flon        | 0.91 | 1.18    | 1.22    | 0.02   | 0.02   | 3.79      | 43.23    |
| Broye       | 0.91 | 21.02   | 22.49   | 0.87   | 0.94   | 6.98      | 8.57     |

**Figure A5.** Monthly and annually aggregated spatial patterns of key fluxes. Q = discharge, P = precipitation, T mean = average temperature, ET/PET= ratio of actual to potential evapotranspiration, ET/P = ration of actual evapotranspiration to precipitation. Qdiff = absolute difference in discharge between base and example SOC increase scenario (MedC\_LowC).

2018

2019

LowC\_vLowC

2020

2021

MedC\_LowC

#### Appendix: A12 690

### Absolute difference in daily discharge at Q2 events Broye Arbogne 1.00 Q base: 77.17 Q base: 70.29 Absolute difference in discharge (m3/s) $^{0.25}_{0.10}$ $^{0.15}_{0.00}$ $^{0.10}_{0.05}$ Q base: 5.12 Q base: 5.94 0.75 0.50 0.25 Petit Glâne Flon d base: 2.52 Q base: 3.21 0.30 0.25 Q base: 11.61

0.20 0.15

2018

2019

2020

2021

vHighC\_MedC

Figure A6. Absolute difference in discharge for peak flow events for SOC scenarios vs. the base scenario. "Q base" denotes the absolute discharge value for each event.

**Figure A7.** Relative difference in PAWC for SOC cs. base scenario for different soil textures. We can see, that PAWC increase is more sensitive to higher sand contents.

**Figure A8.** Difference in number of low flow days for base vs. SOC scenario (example scenario MedC\_lowC) for 6 different optimization runs with the same setting.s

**Table A3.** Default and optimized mHM parameters used in this study. Column "Name (this study)" refers to the names given those parameters in the context of this study for practical reasons.

| Parameter (mHM)                    | Unit                            | Default value | Calibrated value | Name (this study)          |
|------------------------------------|---------------------------------|---------------|------------------|----------------------------|
| canopyInterceptionFactor           | -                               | 0.1500        | 0.2567           |                            |
| snowTreshholdTemperature           | °C                              | 1.0000        | 1.4050           |                            |
| degreeDayFactor_forest             | $\text{m}^{\circ}\text{C}^{-1}$ | 1.5000        | 3.9656           |                            |
| degreeDayFactor_impervious         | $\text{m}^{\circ}\text{C}^{-1}$ | 0.5000        | 0.9776           |                            |
| degreeDayFactor_pervious           | $\text{m}^{\circ}\text{C}^{-1}$ | 0.5000        | 1.9552           |                            |
| increaseDegreeDayFactorByPrecip    | $^{\circ}\mathrm{C}^{-1}$       | 0.5000        | 0.3843           |                            |
| maxDegreeDayFactor_forest          | $\text{m}^{\circ}\text{C}^{-1}$ | 3.0000        | 7.8540           |                            |
| maxDegreeDayFactor_impervious      | $m^{\circ}C^{-1}$               | 3.5000        | 7.9992           |                            |
| maxDegreeDayFactor_pervious        | $m^{\circ}C^{-1}$               | 4.0000        | 7.9497           |                            |
| orgMatterContent_forest            | %                               | 3.4000        | 0.0041           |                            |
| orgMatterContent_impervious        | %                               | 0.1000        | 0.9951           |                            |
| orgMatterContent_pervious          | %                               | 0.6000        | 0.0000           |                            |
| PTF_lower66_5_constant             | -                               | 0.7600        | 0.7627           | $P_{ m constant}$          |
| PTF_lower66_5_clay                 | -                               | 0.000900      | 0.000122         | $P_{ m clay}$              |
| PTF_lower66_5_Db                   | -                               | -0.2640       | -0.3160          | $P_{ m BD}$                |
| PTF_higher66_5_constant            | -                               | 0.8900        | 1.1042           |                            |
| PTF_higher66_5_clay                | -                               | -0.0010       | 0.0048           |                            |
| PTF_higher66_5_Db                  | -                               | -0.3240       | -0.1002          |                            |
| rootFractionCoefficient_forest     | -                               | 0.9700        | 0.9705           |                            |
| rootFractionCoefficient_impervious | -                               | 0.9300        | 0.9850           |                            |
| rootFractionCoefficient_pervious   | -                               | 0.0200        | 0.9753           |                            |
| infiltrationShapeFactor            | -                               | 1.7500        | 2.3038           | e <sub>soil_moisture</sub> |
| rootFractionCoefficient_sand       | -                               | 0.0200        | 0.9753           | $f_{ m sand}$              |
| rootFractionCoefficient_clay       | -                               | 0.0200        | 0.9753           | $f_{ m clay}$              |
| FCmin_glob                         | -                               | 0.1500        | 0.1500           | $	heta_{	ext{min}}$        |
| FCdelta_glob                       | -                               | 0.2500        | 0.2500           | $	heta_{ m global}$        |
| jarvis_sm_threshold_c1             | -                               | 0.5000        | 0.5161           | $T_{ m jarvis}$            |
| imperviousStorageCapacity          | cm                              | 0.5000        | 4.9946           |                            |
| minCorrectionFactorPET             | -                               | 0.9300        | 0.8664           |                            |
| maxCorrectionFactorPET             | -                               | 0.1900        | 0.1598           |                            |
| aspectTresholdPET                  | -                               | 171.00        | 160.22           |                            |
| interflowStorageCapacityFactor     | mm                              | 85.000        | 82.543           |                            |
| interflowRecession_slope           | -                               | 7.0000        | 4.0930           |                            |
| fastInterflowRecession_forest      | -                               | 1.5000        | 1.0324           |                            |
| slowInterflowRecession_Ks          | -                               | 15.0000       | 2.7844           |                            |
| exponentSlowInterflow              | -                               | 0.1250        | 0.2985           |                            |
| rechargeCoefficient                | -                               | 35.000        | 14.867           |                            |
| rechargeFactor_karstic             | _                               | -1.000        | -1.000           |                            |

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
