# Peer review of "From Soil to Stream: Modeling the Catchment-Scale Hydrological Effects of Increased Soil Organic Carbon"

_EGUsphere, 2025_

## Referee Comment (RC1)

Review for "From Soil to Stream: Modeling the Catchment-Scale Hydrological Effects of Increased Soil Organic Carbon" by Heinz et al.

General comments:

In this paper, the authors numerically investigate how increased soil organic carbon (SOC) affects catchment-scale hydrological processes by modifying soil hydraulic properties. Hydrological responses of catchments are an important topic in a changing climate, and the manuscript is well structured and written. However, I am not convinced with the assumption that the authors only consider SOC increases and implement a uniform increase in SOC for the permeable cells across the catchment. This assumption appears unrealistic as SOC can either increase or decrease across the catchment. In addition, I am doubtful about the accuracy of the pedotransfer function linking SOC changes to soil hydraulic properties. See my comments in detail below.

Major comments:

1. As mentioned before. I am not convinced that assuming a uniform increase in SOC across the entire catchment is appropriate. SOC can either increase or decrease across the catchment. I understand that SOC changes can be significant in agricultural areas, whereas SOC changes may be less pronounced in unmanaged forests and grasslands, which constitute important parts of the catchment.
   Reference: Terrer, C., Phillips, R. P., Hungate, B. A., Rosende, J., Pett-Ridge, J., Craig, M. E., ... & Jackson, R. B. (2021). A trade-off between plant and soil carbon storage under elevated $CO_2$. Nature, 591(7851), 599-603.
2. Equations (1-3) and Section 4.2. Given that SOC change effects are reflected by soil hydraulic properties, it is important to estimate these functions for the catchment investigated. If these equations are not suitable for the current catchment, the numerical results will be biased. In addition, I would suggest to discuss why SOC change effects can be adequately captured solely by changes in soil hydraulic properties.
3. Model calibration and evaluation and Figure 5. The numerical model works well in reproducing the discharge. However, it would be better if the authors can provide the comparison between numerical results and measurements for ET, soil water content, and subsurface runoff.
4. Section 4.3 and 4.4. The authors show the difference of ET, soil water content, and subsurface runoff at the grid scale, while the difference of discharge at the catchment scale. Given that the difference of discharge is very small at the catchment scale (Lines 364-366), I would suggest showing the difference of ET, soil water content, and subsurface runoff at the catchment scale too. In addition, the SOC increase seems to have a negligible effect on hydrological processes, considering the uncertainty from assuming the uniform SOC change.

Minor comments:

1. Lines 6-8. The authors only report result at the grid scale, but the title involves the catchment scale. I would suggest mentioning catchment-scale results as well.
2. Section 2.2. The time step is missing.
3. Line 205. Spaces are needed for "$C_{Ksat1}$ to $C_{Ksat6}$ are".
4. Lines 232-233 and Figure 3. 2017 appears dry and hot.
5. Lines 264-268. "selected" to "select".
6. Line 330. "dont" to "do not".
7. Line 332. Suggest removing this sentence.
8. Lines 334-335. Please correct this sentence.
9. Figure 7a. The legend is missing. Change "scenario" to "base scenario" in the legend of Figure 7c.
10. Line 367. Figure 9b is missing.
11. Lines 459-460. Does this mean that we can ignore SOC effects on hydrological processes?
12. Line 467. This is misleading. ET is widely measured by flux towers (see https://fluxnet.org/).
13. Line 619. "takeaways" to "findings".
14. Line 672. The content of Appendix A5 is missing.
15. Lines 785-790. Repeated reference.
16. There are too many figures in the appendix. I would suggest moving some of them to the supporting information.